Investigating the sequence landscape in the Drosophila initiator core promoter element using an enhanced MARZ algorithm

Dresch Jacqueline M.
Conrad Regan D.
Klonaros Daniel
Drewell Robert A. rdrewell@clarku.edu
Biology, Clark University , Worcester, MA , United States
Ettrich Rudi
Electronic publication date: 2023 Jun 22
Publication date: 2023
Volume: 11
Electronic Location ID: e15597
Received 2023 Feb 27; Accepted 2023 May 29
Copyright: © 2023 Dresch et al.
Copyright year: 2023
Copyright holder: Dresch et al.
License: This is an open access article distributed under the terms of the Creative Commons Attribution License, which permits unrestricted use, distribution, reproduction and adaptation in any medium and for any purpose provided that it is properly attributed. For attribution, the original author(s), title, publication source (PeerJ) and either DOI or URL of the article must be cited.
License URL: https://creativecommons.org/licenses/by/4.0/

Keywords: Transcription, DNA binding, Initiator, TATA box, Promoter

Funding: National Institutes of Health GM110571 and GM137250 This work was funded by National Institutes of Health grants (GM110571 and GM137250) to Robert Drewell and Jacqueline Dresch. There was no additional external funding received for this study. The funders had no role in study design, data collection and analysis, decision to publish, or preparation of the manuscript.

==============================
The core promoter elements are important DNA sequences for the regulation of RNA polymerase II transcription in eukaryotic cells. Despite the broad evolutionary conservation of these elements, there is extensive variation in the nucleotide composition of the actual sequences. In this study, we aim to improve our understanding of the complexity of this sequence variation in the TATA box and initiator core promoter elements in Drosophila melanogaster. Using computational approaches, including an enhanced version of our previously developed MARZ algorithm that utilizes gapped nucleotide matrices, several sequence landscape features are uncovered, including an interdependency between the nucleotides in position 2 and 5 in the initiator. Incorporating this information in an expanded MARZ algorithm improves predictive performance for the identification of the initiator element. Overall our results demonstrate the need to carefully consider detailed sequence composition features in core promoter elements in order to make more robust and accurate bioinformatic predictions.

Introduction

Core promoter elements

The promoter is an essential stretch of non-protein coding DNA for the regulation of gene expression (Juven-Gershon et al., 2008; Ohler & Wassarman, 2010). More specifically, RNA polymerase II (RNAPII) core promoters are responsible for the transcriptional initiation of all protein coding genes within eukaryotic cells (Ohler & Wassarman, 2010; Hampsey, 1998; Juven-Gershon & Kadonaga, 2010; Kadonaga, 2012). As it directs the complex, but fundamental, molecular mechanism of gene expression, even a small fluctuation in the sequence of the core promoter, such as a single nucleotide deletion, can impair transcriptional activity (Javahery et al., 1994; Kutach & Kadonaga, 2000) and potentially impact many downstream processes, including causing developmental abnormalities or diseases such as cancer. The critical biological significance of these RNAPII core promoters make it essential to accurately identify and functionally characterize their composition in the genome.

The canonical RNAPII core promoter commonly consists of core promoter elements (CPEs) such as the TATA box (TATA), the initiator (INR), and the downstream promoter element (DPE) (Fig. 1A). The CPEs are known to play an important role in recruitment of RNAPII to actively expressed genes, but have been shown to exhibit some degree of sequence variation and not all are present in every promoter (Luse et al., 2020; Tome, Tippens & Lis, 2018; Vo Ngoc et al., 2017). Nonetheless, the CPEs, along with the molecular machinery and basic functional features of RNAPII, are evolutionarily conserved across a wide selection of eukaryotes, including Homo sapiens and Drosophila melanogaster (Allison et al., 1988; Bailey & Gribskov, 1998; Jin et al., 2006). In both of these species the spatial arrangement of the CPEs in the promoter are shared (Burke & Kadonaga, 1997; Vo Ngoc, Kassavetis & Kadonaga, 2019) and, in a comparative analysis, the only sequence motifs that were found to demonstrate robust conservation were that of the TATA, INR, and DPE (FitzGerald et al., 2006). Despite this overall conservation, all three CPEs exhibit some degree of sequence variation within their 6 bp binding site motif (Fig. 1). The CPEs function in the direct binding of general transcription factors (GTFs) to the core promoter to form a pre-initiation complex (PIC), so that the genomic vicinity can eventually be recognized and bound by RNAPII to initiate transcription (Orphanides, Lagrange & Reinberg, 1996). One such GTF is the TFIID complex, which consists of TATA-binding protein (TBP) and TBP-associated factors (TAFs). TFIID recognizes the INR and TATA regions as binding sites (Juven-Gershon et al., 2008). Overall, INR is found in a higher proportion of promoters in comparison to TATA (Périer et al., 2000), interacts directly with the TAF1 and two subunits of TFIID, and is often spanning the +1 transcription start site (TSS) (Luse et al., 2020; Tome, Tippens & Lis, 2018; Vo Ngoc, Kassavetis & Kadonaga, 2019). However, even in the absence of the TFIID, there are additional INR-binding proteins such as TFII-I and YYI that can direct transcription initiation once the INR has been recognized (Kaufmann et al., 1996). It has been suggested that binding proteins such as TFII-I and YYI are sequence-specific and participate only at a subset of promoters (Butler & Kadonaga, 2002).

Figure 1 Drosophila core promoter elements.

(A) Depiction of the TATA box (TATA), initiator (INR), and downstream promoter element (DPE) as common core promoter elements in the RNAPII core promoter. The consensus sequences for each, along with their usual locations, are included. Notably, the INR encompasses the +1 transcription start site. The consensus sequences are depicted with the code: W = A or T; R = A or G; Y = C or T; B = C, G, or T; V = A, C, or G. (B) Probability matrix for Drosophila TATA core promoter element. (C) Probability matrix for Drosophila INR core promoter element. The TATA and INR probability matrices are derived from the Drosophila core promoter database (Kutach & Kadonaga, 2000) input sequences analyzed in this study (shown in Tables S1 and S2). The sequence logos representing the TATA and INR matrices are depicted in Fig. 2.

Figure 2 MARZ32 models identify key nucleotides in CPEs.

(A) Sequence logo for the 86 Drosophila TATA sequences used in this study. (B) Sequence logo for the 205 Drosophila INR sequences used in this study. Both sequence logos conform to the previously described consensus sequences and highlight the general conservation of the TATA and INR core promoter elements in fly. The total number of times a nucleotide position is considered in the good (green, top line) and poor (red, bottom line) performing MARZ models are shown below the sequence logos for all models and only the models that are six nucleotides in length (6 nt).

Core promoter as a critical developmental regulatory switch

Recent studies have confirmed that the core promoter region is an active regulatory module capable of controlling transcriptional programs in evolutionarily conserved gene regulatory networks during embryonic development (Sloutskin et al., 2021). Live imaging on Drosophila embryos revealed that promoter composition dictates transcriptional bursting linked to alternating active and inactive promoter states (Pimmett et al., 2021). Specifically, TATA-containing promoters exhibit long active states, high rates of polymerase initiation and infrequent inactive states (Pimmett et al., 2021). In contrast, INR is associated with an increased frequency of two distinct inactive states, one of which is connected with promoter-proximal pausing of polymerase (Pimmett et al., 2021). Detailed analysis of over 3,000 synthetic promoters harboring mutations in the CPEs confirms a strong impact on transcriptional output, with a linear combination of the individual CPE motifs largely accounting for the combinatorial impact on core promoter activity (Qi et al., 2022). In our current study, we aim to provide insight into the sequence variation observed within the well-studied TATA and INR CPEs.

Experimental and bioinformatic challenges

To date both experimental and bioinformatic approaches have been used to identify and characterize CPEs within a particular genome. Experimental approaches including in vivo binding studies, such as ChIP-seq to discover CPE locations through protein binding (Mundade et al., 2014), and nuclear run-on to map TSSs and associated CPE sequences (Luse et al., 2020; Tome, Tippens & Lis, 2018) have proven essential to obtain the most informative mapping. Indeed, the base pair precision provided by the RNA run-on experimental approach has indicated that the core INR sequence is a simple 2 bp 5′-CA-3′ motif at position −1 and +1 relative to the TSS, when analyzed at the promoter region of over 78,000 mammalian genes (Tome, Tippens & Lis, 2018). In human promoters, the same 2 bp INR motif emerges across a dataset that contains over 177,000 core promoters, with sequence variation in the extended 6 bp INR sequence correlating with promoter strength (Luse et al., 2020). While such experimental approaches are undoubtedly valuable, they are costly and time consuming to perform as they require exceptional sequencing depth (Luse et al., 2020).

In silico bioinformatic models represent a complementary, more cost effective and rapid approach to identifying CPE sites based on predictions from initial experimental data, but with the potential trade-off that they may be less accurate than experimental techniques. Many current bioinformatic models use position weight matrices (PWMs) populated from characterized CPE sequences, along with software such as PATSER or MAST from the MEME suite (Bailey & Gribskov, 1998; Hertz & Stormo, 1999), to predict the locations of previously uncharacterized CPEs. One limitation to this approach is that simple PWMs operate under the assumption that the appearance of a nucleotide at any particular position is independent of the other positions in the binding site (Schneider & Stephens, 1990; Siddharthan, 2010). This assumption has been argued to provide a valid approximation that allows for the accurate identification of binding sites (Benos, Bulyk & Stormo, 2002). However, other studies have indicated that this assumption may lead to discrepancies between model predictions and experimental data and that including interdependencies between positions provides a more accurate depiction of binding site activity (Bulyk, Johnson & Church, 2002; Man & Stormo, 2001; Segal et al., 2007), TF binding site motifs in complex datasets (Siebert & Söding, 2016) and transcriptional activity of the elements of the downstream core promoter region (Vo Ngoc et al., 2020).

MARZ algorithm as a tool to investigate sequence interdependencies in CPEs

The combinatorial matrix analysis and ranking inspired by zero-knowledge proofs (MARZ) algorithm has been a valuable tool in developing our understanding of the complex interdependencies between nucleotides in binding sites (Dresch et al., 2016; Zellers, Drewell & Dresch, 2015). In contrast to other approaches, MARZ utilizes gapped n-mer models by using a system of k (ignored) and m (considered) nucleotides to create a set of 32 matrix models for a six-base pair long sequence (Table 1). Critically, MARZ enables the identification of interdependencies between any nucleotide positions in the sequences considered (see Methods for full details).

Table 1 MARZ algorithm utilizes 32 different sequence matrix models.

Model number	Gapped n-mer	
0	m	
1	mm	
2	mkm	
3	mmm	
4	mkkm	
5	mkmm	
6	mmkm	
7	mmmm	
8	mkkkm	
9	mkkmm	
10	mkmkm	
11	mkmmm	
12	mmkkm	
13	mmkmm	
14	mmmkm	
15	mmmmm	
16	mkkkkm	
17	mkkkmm	
18	mkkmkm	
19	mkkmmm	
20	mkmkkm	
21	mkmkmm	
22	mkmmkm	
23	mkmmmm	
24	mmkkkm	
25	mmkkmm	
26	mmkmkm	
27	mmkmmm	
28	mmmkkm	
29	mmmkmm	
30	mmmmkm	
31	mmmmmm	
Note:

The first column lists the Model Number for each gapped n-mer and the second column lists the corresponding m/k representation, illustrating which nucleotides are considered (m) and which are ignored (k) when scoring a potential core promoter element sequence.

In this study, we utilize the MARZ algorithm to investigate the TATA and INR CPEs in 205 well-defined, experimentally verified D. melanogaster promoters (Kutach & Kadonaga, 2000). Initial analysis reveals key sequence composition differences between the two CPEs. While the different MARZ models perform in a relatively homogenous manner on TATA, irrespective of the threshold employed, the performance on INR is more complex and detects intriguing sequence features which warrant detailed investigation. Several sequence landscape features, notably an interdependency between the nucleotides in position two and five in INR, are uncovered. Incorporating this information in an expanded MARZ algorithm that includes 64 models demonstrates that improved predictive performance for the identification of the INR CPE can be obtained. Overall, the results demonstrate the need to carefully consider sequence composition features in CPEs, particularly in the case of INR, in order to improve the accuracy of bioinformatic predictions.

Methods

Input data sources

As inputs for the MARZ-based analysis of the D. melanogaster TATA-box (TATA) and initiator (INR) core promoter elements (CPEs), aligned sequences were curated from the Drosophila core promoter database (https://labs.biology.ucsd.edu/Kadonaga/DCPD.htm). The database contains 205 experimentally verified fly promoters (Kutach & Kadonaga, 2000), including 86 six bp annotated TATA sequences (raw sequence data shown in Table S1) and 205 six bp annotated INR sequences (raw sequence data shown in Table S2). The second input was bioinformatically predicted D. melanogaster promoter sequences curated from the eukaryotic promoter database (Périer et al., 2000). In total, 2,326 promoters with a predicted TATA CPE and 8,169 promoters with a predicted INR CPE were used. These promoter regions were trimmed to 100 bp sequences (−49 bp to +50 bp relative to annotated transcription start site). Finally, background nucleotide frequencies were calculated from the entire D. melanogaster genome (A/T 0.6 and G/C 0.4). The work flow for these inputs is shown in Fig. 3.

Figure 3 Work flowchart of the MARZ algorithm implementation and associated analyses on promoter sequences.

The complete work flow is depicted from top to bottom, as the three boxes at the top (orange) illustrate the three inputs to the MARZ algorithm and the large box at the bottom (green) illustrates the values the MARZ algorithm outputs and the various analyses performed on the results.

MARZ approach and outputs

The combinatorial matrix analysis and ranking inspired by zero-knowledge proofs (MARZ) algorithm combinatorially analyzes gapped n-mer matrices, a system that allows for the analysis of dependency and independency between nucleotide positions. Each model consists of a string of ks and ms, where a k indicates a position that is ignored, and an m indicates a considered position (Zellers, Drewell & Dresch, 2015).

These 32 models were used to generate 32 different position weight matrices (PWMs) from the aligned TATA and INR sequences, independently. The MARZ algorithm was run with each of these 32 TATA or INR models with five thresholds of ascending selectivity (0, 0.25, 0.50, 0.75, and 1) on the respective TATA or INR promoter datasets. When the weight score of an input sequence scores higher than the threshold, it is designated as a binding site (Zellers, Drewell & Dresch, 2015).

The algorithm uses the RZ scoring system to score the performance of each model at each threshold. This scoring system reflects the ability of a model to distinguish between the identification of binding sites in real promoter sequences and scrambled sequences. Note that the scrambled sequences are created using the original sequences to maintain the same nucleotide composition and avoid any bias toward one nucleotide over another when scoring. An RZ score of 0 indicates that the algorithm only identifies binding sites in the scrambled sequences, while an RZ score of 1 indicates that sites were only found in the real sequences. A score of 0.5 indicates that an equal number of predicted binding sites were identified in the scrambled and real sequences. Thus, RZ scores for each model at each scoring threshold range from 0 to 1. The MARZ algorithm outputs RZ scores for each model, at all five designated thresholds, for both the TATA (Fig. 4) and INR (Fig. 5) CPEs (see Fig. 3 for complete workflow).

Figure 4 Heat map of MARZ scores for Drosophila TATA containing core promoters.

Heat map of MARZ32 results for TATA sequences. A blue color indicates a high RZ score and a red color indicates a low RZ score, with intensity increasing as the RZ scores deviate further from the median RZ score. For simplicity, RZ scores are multiplied by one hundred to produce a 0–100 scale.

Figure 5 Heat map of MARZ scores for Drosophila INR containing core promoters.

Heat map of MARZ32 results for INR sequences. A blue color indicates a high RZ score and a red color indicates a low RZ score, with intensity increasing as the RZ scores deviate further from the median RZ score. For simplicity, RZ scores are multiplied by one hundred to produce a 0–100 scale.

RZ score analysis

The models with the highest (good performing) and lowest (poor performing) RZ scores were identified for each CPE at each of the five thresholds. The RZ score cut-off values were chosen to identify a median of three good and three poor models per threshold, although the distribution of scores at some thresholds resulted in more or fewer models being categorized as good or poor performers. The ID numbers of good and poor performing models were recorded for TATA (Table S3) and INR (Table S4). These ID numbers were then used to identify the actual m and k nucleotides that are used in the good and poor performing models. The frequency that each position (1 through 6) appears in the model as an m (considered) nucleotide was summed, accounting for the sliding window used by the various MARZ models, at each threshold and as a total for each threshold level (Fig. S1). The totals of considered nucleotides across all thresholds were calculated, as well as the totals of only the six nucleotide-long models, thus excluding potential bias from the sliding window in the shorter models.

Sequence logos

Initial sequence logos representing the TATA and INR CPEs identified in the 205 promoters from the Drosophila core promoter database were generated from the PWMs shown in Tables S1 and S2, using the Berkeley weblogo 2.8.2 site (https://weblogo.berkeley.edu/logo.cgi) (Crooks et al., 2004). Additional sequence logos were created using this same software to represent the good and poor performing models (identified as described in the previous section). The sequence data used to create these was modified to reflect the k and m representation of any model. In the case of a k (ignored) nucleotide, that position was changed to N (so that no nucleotide would be accounted for in that position). For example, the sequence TCAGTG, if modified to model 25 (mmkkmm), would become TCNNTG. This method also accounted for the sliding window of some of the models that are shorter than six nucleotides in length by placing an N for nucleotides that are not included. Thus, sequence logos were created by combining the modified sequences of nucleotides and Ns for all models of a performance level and threshold and inputting these compilations of sequences into the sequence logo software.

Modified MARZ

To help eliminate potential bias in the start (position 1) and end (position 6) positions of any model, the MARZ algorithm was modified to allow a k (excluded) nucleotide to exist in these starting and ending positions. This generates a full set of 64 six nucleotide long gapped n-mer models, with new model numbers which range from 0 to 63 (shown in Table 2) for binary conversion.

Table 2 Expanded MARZ algorithm utilizes 64 different sequence matrix models.

Type ID	Gapped n-mer	Type ID	Gapped n-mer	Type ID	Gapped n-mer	Type ID	Gapped n-mer	
0	kkkkkk	16	kmkkkk	32	mkkkkk	48	mmkkkk	
1	kkkkkm	17	kmkkkm	33	mkkkkm	49	mmkkkm	
2	kkkkmk	18	kmkkmk	34	mkkkmk	50	mmkkmk	
3	kkkkmm	19	kmkkmm	35	mkkkmm	51	mmkkmm	
4	kkkmkk	20	kmkmkk	36	mkkmkk	52	mmkmkk	
5	kkkmkm	21	kmkmkm	37	mkkmkm	53	mmkmkm	
6	kkkmmk	22	kmkmmk	38	mkkmmk	54	mmkmmk	
7	kkkmmm	23	kmkmmm	39	mkkmmm	55	mmkmmm	
8	kkmkkk	24	kmmkkk	40	mkmkkk	56	mmmkkk	
9	kkmkkm	25	kmmkkm	41	mkmkkm	57	mmmkkm	
10	kkmkmk	26	kmmkmk	42	mkmkmk	58	mmmkmk	
11	kkmkmm	27	kmmkmm	43	mkmkmm	59	mmmkmm	
12	kkmmkk	28	kmmmkk	44	mkmmkk	60	mmmmkk	
13	kkmmkm	29	kmmmkm	45	mkmmkm	61	mmmmkm	
14	kkmmmk	30	kmmmmk	46	mkmmmk	62	mmmmmk	
15	kkmmmm	31	kmmmmm	47	mkmmmm	63	mmmmmm	
Note:

The first column lists the Model Number for each gapped n-mer and the second column lists the corresponding m/k representation, illustrating which nucleotides are considered (m) and which are ignored (k) when scoring a potential core promoter element sequence.

MAST bioinformatic predictions

The motif alignment and search tool (MAST) is a part of the MEME Suite toolbox that can search DNA sequences for a defined motif (Bailey & Gribskov, 1998). The MAST software requires the search motif to be in MEME Motif Format and searches through a user-provided sequence database, which in this case consisted of the 205 promoter sequences from the Drosophila core promoter database (Kutach & Kadonaga, 2000) described above. All 205 sequences provided were 92 bp in length and MAST successfully identified one high scoring INR CPE in each promoter sequence. For all MAST searches the background nucleotide frequencies was set to A/T 0.6 and G/C 0.4, reflecting the overall nucleotide composition of the Drosophila genome. The purpose of these background frequencies is to normalize for any bias in the distribution of individual nucleotides within the genome.

MAST was run using PWMs created from the 64 six nucleotide models in the modified MARZ algorithm described in the previous section. The p-value that was used as a threshold in MAST was determined for each model by maximizing the positive predictive value (PPV) shown in the equation below:

PPV=NumberoftruepositivesNumberoftruepositives+Numberoffalsepositives

A true positive is defined as a MAST-identified hit that has at least one nucleotide that falls within the identified INR CPE in the 92 bp promoter region. A false positive is defined as a MAST-identified hit that does not fall within the identified INR.

To determine the background level of hits, 100 scrambles of each of the 205 promoter sequences were created to serve as search sets on MAST. MAST was then run with each of the 64 MARZ model PWMs on all 205 original (real) sequences as well as their associated 100 scrambles using the calculated p-value that maximized the PPV for each independent model. The average number of hits on the 100 scrambles of each sequence, the total number of hits (true positives and false positives) on the real sequence, and the number of true positive hits on the real sequence were recorded. For each of the 205 promoter sequences, a true hit ratio was calculated using the following equation:

Truehitratio=TruepositivehitsonrealsequenceAllhitsonrealsequence−Averagenumberofhitsonscrambledsequences

The true hit ratios across all 205 promoter sequences were averaged to achieve a mean true hit ratio for each model that reflects its ability to accurately determine true positive hits on the real sequence.

Leave-parts-out analysis

To address the potential redundancy that may arise from using a relatively small number of sequences, namely the 205 defined promoters in the Drosophila Core Promoter Database, to both create PWMs and to use as a search set, leave-parts-out analysis was undertaken. A subset of 102 promoter sequences (representing approximately 50% of the dataset) was randomly selected using the random number generator in MATLAB. Each subset of 102 sequences was used to create a set of 64 PWMs, one for each six nucleotide modified MARZ model, and the remaining 103 sequences were used as the search set. The p-value used in all trials of the leave-parts-out analysis was 0.006, which was the median p-value of those previously determined to maximize the PPV of the 16 six nucleotide models described in the original MARZ algorithm (Zellers, Drewell & Dresch, 2015). MAST was run on all 64 nucleotide models for the 103 real sequences in the search set and the 100 scrambles of each, and the mean true hit ratio was determined for each of the 64 models. This process was undertaken a total of 50 times (5 × 10 sets of trials), resulting in 50 random, independent trials of the leave-parts-out-analysis.

Pearson correlation calculations

The Pearson correlation was calculated between each set of 10 trials using the MATLAB command corr(X). The variable X was a matrix, with each column containing the average mean true hit ratios for the 64 models. Thus, Pearson correlations between each set of 10 trials were calculated to confirm the results were consistent across all trials. The correlations between all sets fell in a range between 0.87 and 0.97, indicating that the presence of outlying data sets was unlikely.

Nucleotide interdependency probabilities

A 4 × 6 position probability matrix was constructed from the 205 six bp annotated INR sequences. The matrix contains the frequency of each nucleotide at each position in the INR site, divided by the total number of sequences (205). For hypothesis testing, we used these probabilities to generate 100,000 sets of 205 random sequences to ensure that the nucleotide composition of each set was comparable to the original set of INR sequences. We then performed a two-tailed test on each pair of positions and nucleotide pairs (15 × 16 = 240 total tests) with the null hypotheses that nucleotides in each pair of positions are independent of one another. For example, to test that the occurrence of an A in position 2 and a G in position 5 are independent of one another, we calculated the p-value (i.e., the proportion of the 100,000 sets of random sequences that contained an A in position 2 and a G in position 5 at a frequency equal to or above that found for the 205 six bp annotated INR sequences) using an expected frequency = 205 × prob(A in pos 2) × prob(G in pos 5). Hypotheses tests were conducted using significance values of α = 0.05, α = 0.01, and α = 0.001. One-tailed tests were also performed for each pair of positions and nucleotide pairs in the INR sequences to test for over/under-enrichment. At a significance level of α = 0.001, the only over-enriched pair found was an A at position 2 and G at position 5 and the only under-enriched pair found was a T at position 5 and an A at position 6.

Results and discussion

MARZ algorithm approach

The unbiased, systematically constructed set of 32 matrices (Table 1) of the MARZ algorithm (Zellers, Drewell & Dresch, 2015) were utilized to investigate the six nucleotide long TATA-box (TATA) and initiator (INR) core promoter element (CPE) sequences in Drosophila melanogaster. The 32 n-mer models used in this study range in length from one to six nucleotides long and all start and end with an m (considered) nucleotide (Table 1). The simplest matrix, m, is generated from a traditional mononucleotide model in which each nucleotide is considered independently. When applied to the TATA and INR sequences, which are six nucleotides long, this creates six frames. A dinucleotide model, mm, considers two adjacent nucleotides and an n-mer model considers n contiguous nucleotides in each frame. In addition to implementing these simple models, our approach examines all possible gapped n-mers with up to a six nucleotide frame size (Zellers, Drewell & Dresch, 2015). When scoring a potential binding site, the gapped n-mer matrices only consider a subset of nucleotides (m) across any given frame and ignore the other nucleotides (k).

The MARZ algorithm was run with these 32 models using five threshold positions: 0, 0.25, 0.5, 0.75, and 1. These threshold positions are used by MARZ to dynamically calculate threshold values (see Methods for full details). Briefly, at the 0.75 threshold position, MARZ calculates the highest threshold value at which sequences in the 75th percentile of the experimentally determined sequence are identified as binding sites (Zellers, Drewell & Dresch, 2015). The threshold values are used to determine whether a potential binding site has scored high enough to be defined as a predicted binding site; if a sequence has a weight score greater than the threshold value, it is designated as a binding site. Accordingly, a high threshold position only permits the prediction of binding sites with strong similarity to the sequence with the highest weight score, while a low threshold position allows for the prediction of both weak and strong binding sites (Dresch et al., 2016). The MARZ algorithm produced RZ scores for each of the 32 models, at all five designated thresholds, for both the INR and TATA sequences (complete workflow is depicted in Fig. 3).

MARZ32 identifies key sequence landscape features in CPEs

The RZ score results for TATA (Fig. 4) and INR (Fig. 5) reveal some distinct sequence composition features for the Drosophila CPEs. For TATA, the overall performance across the 32 models is markedly consistent. Almost all models generate scores in the 0.6–0.8 range irrespective of the threshold (Fig. 4). This uniformity likely results from the overall lack of nucleotide variation found in the annotated TATA sequences from the Drosophila Core Promoter Database (Fig. 1B). However, models 8 (mkkkm), 16 (mkkkkm) and 24 (mmkkkm) clearly under-perform relative to all other models. As all three of these models have ignored (k) positions in their central region this indicates that the nucleotides in positions 2, 3, 4 and 5 may be important for the binding site. This conclusion is supported once more by the very strong consensus displayed in nucleotide positions 2–5 in the TATA CPEs, with a weaker consensus in positions 1 and 6 (Fig. 1B).

In contrast to TATA, the RZ score heat map for INR is much more heterogenous (Fig. 5). This represents a more complex sequence landscape in the nucleotide positions within INR, which is supported by the variance seen at many positions in the annotated INR CPEs from the Drosophila core promoter database (Fig. 1C). To investigate this landscape in more detail, we identified the good and poor performing models at each threshold for each CPE. The cut-off values were chosen to ideally specify three models at each performance level, although the distribution of scores sometimes resulted in more or less models being counted. The ID numbers of models that were categorized as good or poor performers were tabulated for TATA (Table S3) and INR (Table S4) and the composition of m and k nucleotides that are used in the models noted. Finally, the number of times each position (1 through 6) appeared in the model as a m (considered) nucleotide was totaled, accounting for the sliding window used by MARZ, at each threshold (Fig. S1).

Using this scoring metric, the considered nucleotide positions are broadly distributed across TATA, irrespective of whether we analyze all models or just the models that contain six nucleotides (Fig. 2A). However, at INR the center four positions (2, 3, 4 and 5—corresponding to the −1, +1, +2 and +3 base pair positions relative to the TSS) are enriched in good performing models and excluded in poor performing models (Fig. 2B). This is particularly clear when the analysis is restricted to just the six nucleotide-containing models, with a strong enrichment for positions 2 (−1 relative to TSS) and 5 (+3 relative to TSS) in good performing models and a contrasting enrichment for positions 3 and 4 in poor performing models (Fig. 2B). At the 0.25 threshold, positions 3 and 4 are excluded from all good performing models, while positions 2 and 5 are excluded from all the poor performing models. This result suggests that the ability to bioinformatically predict INR CPEs may be enhanced by the selective inclusion (at position 2 and 5) and exclusion (at position 3 and 4) of sequence data at key positions within the CPE. The appearance of nucleotide positions 1 and 6 in both good and poor models is something of an artifact, as the window for every model in the MARZ32 algorithm begins and ends with a considered nucleotide.

Ability to predict INR is impacted by the nucleotides considered

To further investigate the sequence landscape for the INR CPE and to address the potential bias at nucleotide positions 1 and 6 in the MARZ32 models, we generated a full set of 64 six nucleotide long gapped n-mer models (Table 2). The predictive performance for all 64 models was compared using the commonly employed MAST bioinformatic algorithm (Bailey & Gribskov, 1998) to calculate a true hit ratio for successfully identifying the INR CPE in a promoter region over 50 independent trials (see Methods for details) (Table S5). For each MAST trial, the models that had ratio scores performing in the top and bottom 5% of all 64 models (which equates to three models in each category) were recorded. A total of 25 models appear in the top 5% of at least one trial, while a total of 30 models appear in the bottom 5% of at least one trial (Table S6). Out of those, 11 are unique to the top 5% and 16 are unique to the bottom 5%. There are also 14 models that appear at least once in the top 5% and at least once in the bottom 5% (Table S6). Several of the models have a large distribution of mean true hit ratios with many outliers, which may reflect the low number of considered (m) nucleotides in these models.

The three best performing models overall were 58 (mmmkmk), 55 (mmkmmm), and 54 (mmkmmk) (Fig. 6A). Notably, the nucleotides in positions 2 and 5 are included in all three of these top performing models, while positions 3, 4, and 6 are excluded from at least one of the models. The three worst performing models were 2 (kkkkmk), 52 (mmkmkk), and 37 (mkkmkm) (Fig. 6A). Nucleotide positions 2 and 5 are excluded in two of these models, and position 3 is excluded in all three of them. It is important to note that although model 2 does include position 5, all the remaining positions are k (ignored) nucleotides, which likely contributes to its overall poor performance. Ranking all the models by average mean true hit ratio over the 50 trials reveals that many of the models that perform in the top 5% also rank highly for their average mean true hit ratio across all 50 trials (Fig. 6B). Conversely, many of the models that perform in the bottom 5% have low average mean true hit ratios (Fig. 6B).

Figure 6 MARZ64 models display varied INR predictive performance.

(A) Boxplot of mean true hit ratio results, showing data for the 64 models from all 50 trials in order of Model Number. The models appearing most often in the top (green, marked with an asterisk) and bottom (red) 5% of performers for each trial are indicated. (B) Ranked boxplot of mean true hit ratio results, showing data for the 64 models from all 50 trials in descending order of average mean true hit ratio. The models that appear at least once in the top 5% of performers for one trial are indicated in green (marked with an asterisk), those that appear at least once in the bottom 5% of performers for one trial are in red, and those that appear at least once in the top or bottom 5% of performers in at least one trial each are indicated in blue. Note that a model cannot appear in the top and bottom 5% for one trial simultaneously.

In an effort to identify the nucleotides that are contributing to the predictive performance of the top and bottom 5% models, the number of times that an m (considered) nucleotide appeared in each position was recorded for each trial. These numbers were totaled across all 50 trials for each model, then averaged to determine how frequently each position appeared. Strikingly, nucleotide positions 2 and 5 were once more prominent, with position 5 appearing on average 2.54 out of a possible 3 times in the top performing 5% of models and position 2 appearing 2.48 times (Fig. 7). The remaining positions 1, 3, 4, and 6 appeared on average 1.98, 1.7, 2.2, and 1.52 times, respectively in the top 5% performing models (Fig. 7). At all positions within the INR CPE, the frequency of a considered nucleotide was higher in the top 5% models (shown in light grey in Fig. 7) when compared to the bottom 5% models (shown in dark grey in Fig. 7), indicating that a lower number of considered nucleotides is likely reducing the predictive performance of the bottom 5% models. This phenomenon is particularly clear at nucleotide position 3, which appears, on average, only 0.56 times in the bottom 5% (Fig. 7). However, position 3 also has one of the lowest inclusion frequencies amongst the top 5% models, which may simply be a result of the relatively strong consensus for an A nucleotide in this position in the INR CPE.

Figure 7 INR nucleotide position inclusion impacts MARZ64 model performance.

Bar graphs showing the average number of times a m (included) nucleotide appears in the top 5% (light grey) and bottom 5% (dark grey) performing models in the MAST trials. The x axis depicts nucleotide position, ranging from 1–6, in the INR CPE and the y axis shows average number of appearances, ranging from 0–3.

Interdependency at nucleotide positions 2 and 5 in INR

To directly investigate the potential importance of nucleotide positions 2 and 5 in INR we also performed a two-tailed hypothesis test on the 205 annotated INR CPEs. The test was performed on all position pairs (15 total) and all nucleotide pairs (FitzGerald et al., 2006). From this analysis, it is clear that some positions show enrichment for specific nucleotide pairs when compared to other positions (Fig. 8). For example, interdependency appears to be enriched at position 5 (+3 relative to TSS), accounting for 11 out of the 20 total nucleotide pairs showing any significant dependence and includes all four pairs showing significant dependence at α = 0.01 (Fig. 8). Notably, the only nucleotide pair showing significant dependence at the most stringent α = 0.001 level was also found in positions 2 (−1 relative to TSS) and 5 (+3 relative to TSS), where an A in position 2 and a G in position 5 was found to be statistically over-enriched (1-sided test, p < 0.001). These observations are consistent with our MARZ results and emphasize the importance of including information on positions 2 and 5 in bioinformatic predictions of the INR CPE.

Figure 8 Nucleotide interdependency in Drosophila INR binding sites.

Heatmap of p-values corresponds to the two-tailed hypothesis testing performed on each possible pair of nucleotides at each pair of positions within the 6 bp INR sequences. All highlighted values (green, yellow and red) were significant at the a = 0.05 significance level, those highlighted in yellow and red were also found to be significant at the a = 0.01 significance level, and the value in red is the only one found to be significant at the a = 0.001 significance level.

Conclusions

Our analysis uncovers important nucleotide composition differences between the INR and TATA CPEs in Drosophila. The sequence landscape at TATA is relatively homogenous, as evidenced by the strong consensus 5′-TATAAA-3′ at this CPE (Fig. 2A) (Kadonaga, 2012; Vo Ngoc et al., 2017; Vo Ngoc, Kassavetis & Kadonaga, 2019). As a result, the distinct MARZ models all perform in a similar manner on TATA (Fig. 4). In contrast, the MARZ models perform in a much less uniform manner on the INR CPE (Fig. 5), reflecting the diversity of nucleotide composition in this CPE (Fig. 2B). Of particular importance for the bioinformatic predictive performance of the distinct models are the nucleotides in position 2 (−1 relative to TSS) and 5 (+3 relative to TSS) of INR. Including these positions consistently enhances the predictive ability, while excluding them diminishes the accuracy of the predictions (Figs. 6 and 7). Furthermore, our analysis reveals several interdependencies between nucleotides in INR sequences, with the strongest example occurring in position 2 and 5 (Fig. 8). Such interdependencies have been previously characterized in transcription factor binding sites and shown to be important for improving bioinformatic predictions (Dresch et al., 2016; Gershenzon, Stormo & Ioshikhes, 2005; Mathelier & Wasserman, 2013; Weirauch et al., 2013).

Point mutations in any of the six INR sequence positions have been shown to impact transcriptional output (Qi et al., 2022), indicating a potential combinatorial role for all of these nucleotides in mediating Pol II binding. Additionally, previous studies utilizing a combination of bioinformatic algorithms and experimental approaches have confirmed the functional importance of the 5′-TCAGTT-3′ INR in Drosophila melanogaster promoters (Schor et al., 2017) and its evolutionary conservation in the genomes of other Drosophila species (Rach et al., 2009). In contrast, recent high resolution, deep sequencing RNA run-on experiments in mammalian systems have demonstrated a strong enrichment for a minimal 2 bp core INR sequence of 5′-CA-3′ at position −1 and +1 relative to the TSS (Luse et al., 2020; Tome, Tippens & Lis, 2018). We believe that our analysis in this study is significant as the MARZ algorithm captures both of these important properties of the sequence landscape by consistently identifying the nucleotides in position 2 (−1 relative to TSS) and 5 (+3 relative to TSS) as critical to the Drosophila INR element. While the A nucleotide in position 3 (+1 relative to TSS) remains relatively invariant in our dataset (in agreement with the mammalian studies), it contributes little to the predictive ability of the MARZ algorithm. This is not the case for positions 2 and 5, indicating that a functional interplay between the nucleotides at these key positions may mediate INR activity at the 205 Drosophila promoters we examined. In future studies, targeted combinatorial mutational analysis of these critical interdependent nucleotide positions will enable a rigorous assessment of their functional activity. In addition, expanding the application of the MARZ algorithm to larger datasets in a wider range of species will reveal if such interdependencies persist in the INR in promoters of different strength. Overall the results in this current study improve our understanding of the sequence landscape in the Drosophila CPEs and demonstrate the need to carefully consider these features to increase the accuracy of computational predictions of the genomic locations and nucleotide composition of the INR element.

Supplemental Information

Supplemental Information 1 Raw sequence data for Drosophila TATA core promoter elements.

The 86 annotated TATA CPE raw data sequences from the Drosophila Core Promoter Database (Kutach & Kadonaga, 2000) are shown, along with corresponding frequency, probability and position weight matrices. A +1 pseudocount is added to eliminate the appearance of a zero frequency at any position.

Click here for additional data file.

Supplemental Information 2 Raw sequence data for Drosophila INR core promoter elements.

The 205 annotated INR CPE raw data sequences from the Drosophila Core Promoter Database (Kutach & Kadonaga, 2000) are shown, along with corresponding frequency, probability and position weight matrices. A +1 pseudocount is added to eliminate the appearance of a zero frequency at any position.

Click here for additional data file.

Supplemental Information 3 Qualitative analysis of relative performance of MARZ models for Drosophila TATA core promoter element.

Models were identified as Good (green) or Poor (red) performers based on RZ score. Cut-off values to determine the qualification of an RZ score was determined at each threshold (0, 0.25, 0.50, 0.75 and 1). These cut-off RZ values were chosen to ideally specify three Good and three Poor models at each threshold, although the distribution of scores sometimes resulted in more or fewer models being counted. The Model Number and corresponding gapped models are shown (m = considered nucleotide position, k = ignored nucleotide position). The frequency of the considered nucleotide positions across the sliding window employed by each model is depicted in the final column.

Click here for additional data file.

Supplemental Information 4 Qualitative analysis of relative performance of MARZ models for Drosophila INR core promoter element.

Models were identified as Good (green) or Poor (red) performers based on RZ score. Cut-off values to determine the qualification of an RZ score was determined at each threshold (0, 0.25, 0.50, 0.75 and 1). These cut-off RZ values were chosen to ideally specify three Good and three Poor models at each threshold, although the distribution of scores sometimes resulted in more or fewer models being counted. The Model Number and corresponding gapped models are shown (m = considered nucleotide position, k = ignored nucleotide position). The frequency of the considered nucleotide positions across the sliding window employed by each model is depicted in the final column.

Click here for additional data file.

Supplemental Information 5 Mean True Hit Ratios for all 64 Models.

The first column lists the Model Number for each gapped n-mer, the second column lists the corresponding m/k representation, and the third column lists the average Mean True Hit Ratio over the 50 trials. The remaining columns give the Mean True Hit Ratio values obtained from each of the 50 individual trials across all 64 models.

Click here for additional data file.

Supplemental Information 6 Models that appear in the top and bottom 5% of at least one of the 50 MAST trials.

(A) Top 5% models. The first column is the Model Number, which range from 0 to 63. The second column is the number of times that model appeared in the top 5% of the trials; this number can range from 1 (if it appeared in the top 5% of one of the 50 trials) to 50 (if it appeared in the top 5% of every trial). The third column is the k/m representation of each model. Models that appear in the top 5% of trials are shown in green. If a model also appears at least once in the bottom 5% of one of the trials it is shown in blue. (B) Bottom 5% models. Models that are unique to the bottom 5% are shown in orange. Models that also appear in the top 5% of one of the trials are shown in blue. Both tables are in descending order of average mean true hit ratio.

Click here for additional data file.

Supplemental Information 7 Quantification of considered nucleotides in MARZ models.

The method of totaling considered nucleotides, using data for INR threshold 0 poor performance (models 6, 10 and 20) as an example. If a considered nucleotide (red) appears in the sliding window, then a 1 is added to the total for that nucleotide position. The total number of times a position is considered is then summed for all models.

Click here for additional data file.

Abbreviations

RNAPII RNA polymerase II

CPE Core Promoter Element

TATA TATA-box

INR Initiator

DPE Downstream Promoter Element

GTF General Transcription Factor

PIC Pre-Initiation Complex

TBP TATA-binding protein

TAF TBP-associated factor

TSS Transcription Start Site

PWM Position Weight Matrix

MARZ Matrix Analysis and Ranking inspired by Zero-knowledge proofs

MAST Motif Alignment and Search Tool

Additional Information and Declarations

Competing Interests

Author Contributions

Data Availability

The authors declare that they have no competing interests.

Jacqueline M. Dresch conceived and designed the experiments, analyzed the data, prepared figures and/or tables, authored or reviewed drafts of the article, and approved the final draft.

Regan D. Conrad performed the experiments, analyzed the data, prepared figures and/or tables, authored or reviewed drafts of the article, and approved the final draft.

Daniel Klonaros performed the experiments, analyzed the data, authored or reviewed drafts of the article, and approved the final draft.

Robert A. Drewell conceived and designed the experiments, analyzed the data, prepared figures and/or tables, authored or reviewed drafts of the article, and approved the final draft.

The following information was supplied regarding data availability:

The raw sequence data are available in the Supplemental Files.

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
