# Peer review of "Investigating the sequence landscape in the Drosophila initiator core promoter element using an enhanced MARZ algorithm"

_PeerJ, doi:10.7717/peerj.15597_

## Round 0.1 · original submission · Minor Revisions

The reviewers agreed that your manuscript is generally well-written, logically clear and good to understand. There is also agreement that the methods used in the study are in principle appropriate to address the intended questions. Nevertheless, there are a few issue the reviewers raised that should be addressed prior to publication:

For the observed interdependency between nucleotide positions 2 and 5 in the initiator one reviewer remarks a potential bias towards specific bases. Please address or comment on this remark in your manuscript. Please also follow the advice to add in the conclusion section detail about how other algorithms have reported similar findings and why is the finding of MARZ significantly different

Reviewer 2 also has suggestions that I strongly urge you to consider, such as the deletion of Figure panels 3c-d and the inclusion of the original Mean True Hit Ratios for all the 64 models in the supplementary data.

Both reviewers have a couple of suggestions regarding verbiage/formatting/labels/consistency/typos that might improve the paper significantly, please read both reviews carefully and consider implementation.

Reviewer 1 ·

Basic reporting

While the english is clear in most aspects, there are several grammatical errors and spelling mistakes that need to be taken care of. Please see some below
- Please add a comma on line 31 after ‘In this study’
- Line 70 : Please change subsets to subset
- Line 93: Please change At human to in human promoters
- Line 102 software to softwares
- Line 419: Please change ‘show’ to ‘shown’.

Experimental design

The manuscript by Darusch et al, uses MARZ algorithm to investigate the sequences of Drosophila cis- elements. The authors apart from uncovering cis-acting features, opine an interdependency of nucleotide positions with in the initiator element. The manuscript uses Drosophila promoter sequences that were experimentally verified as well as predicted promoters from EPD. In addition, they use genomic sequences as the background. For accurate predictions, factors like k-mer size, selection of negative/scrambled sequences etc are some standards used in this approach, which the authors have used in this study.

Validity of the findings

selecting scrambled or random sequences from the same genome may limit the models and may end up finding simple features, which may not be biological relevant and may be biased towards specific bases. The authors observe an interdependency between nucleotide position 2 and 5 in the initiator. Are there any repeating bases at these positions that the tool is biased for. For instance, some prediction tools show a bias for A or T nucleotides. Do the authors see something similar?

In conclusion section, please add detail about how other algorithms have reported similar findings and why is the finding of MARZ significantly different.

Additional comments

Please add more description to the figure titles. As of now, the titles do not explain much. Please add more description for the reader.

Reviewer 2 ·

Basic reporting

no comment

Experimental design

no comment

Validity of the findings

The authors showed Mean True Hit Ratios for all the 64 models with some of those models having huge within group variations. The authors should include the original Mean True Hit Ratios for all the 64 models in the supplementary data.

Additional comments

In this manuscript, Dresch and coauthors performed extensive computational analysis of nucleotide composition in the core promoter elements of Drosophila melanogaster using both experimentally verified and predicted sequences and found that the initiator core sequence (INR) has more variability compared with the uniformly consistent TATA box sequence. Interestingly, the authors uncovered the interdependency between the positions 2 and 5 in the INR and indicated its importance in determining predictive performance of bioinformatics algorithms. Overall, the manuscript is well-written, logically clear and easy to understand. The methods used in the study are appropriate to address the intended questions. However, there are some minor issues and errors that can be addressed.
The following comments may help improve the manuscript.
1. It is pleasant to see that the authors used several different subtitles in the Introduction section to separate and highlight different topics in order to help the readers understand the background of the study. However, the introduction is too long in length and should be greatly shortened. Some content between lines 52-71 may be moved to beginning of the Results section. Also, the content between lines 113-127 has obvious overlap with the content shown from line 154.

2. In line 205, the phrase “64 6 nucleotide long gapped …” may look confusing to readers. It is suggested to use hyphens between the words 6, nucleotide, and long (such as “64 6-nucleotide-long gapped …”) to eliminate the confusion. There are many other similar phrases in the manuscript.

3. It is suggested to remove Fig 3c-d as the frequencies of the models with the indicated positions considered are too low to be statistically convincible. Meanwhile, the point the authors intended to make is already evident in Fig 3b.

4. The sentences in lines 360-362 and lines 370-372 are unnecessarily repeated.

5. The authors showed Mean True Hit Ratios for all the 64 models with some of those models having huge within group variations. The authors should include the original Mean True Hit Ratios for all the 64 models in the supplementary data.

6. In line 388, the authors claimed that “position 3 also has the lowest inclusion frequency amongst the top 5% models”. Actually, it is the position 6 that has the lowest inclusion frequency according to Fig 5.

7. In line 412, “nucleotides in position 2 (-1 and 5 of INR” has a typo.

8. In the different Tables, the authors either used Type ID or Model # to refer to different models, while in the manuscript, the authors only used model number to refer to specific model. This causes confusion to readers. The authors should use consistent labels to refer to specific models.

---

## Round 0.2 · accepted · Accept

I thoroughly read your revised version and concluded that you did address all concerns and remarks raised by reviewers as well as myself to my fullest satisfaction. As this was a minor revision and you did satisfactorily address all comments I did not re-invite the previous reviewers. I am happy with the current version and consider the manuscript is ready for publication.